# The Predictive Value of Perioperative Inflammatory Indexes in Major Arterial Surgical Revascularization from Leriche Syndrome

**DOI:** 10.3390/jcm13216338

**Published:** 2024-10-23

**Authors:** Anca Drăgan, Adrian Ştefan Drăgan, Ovidiu Ştiru

**Affiliations:** 1Department of Cardiovascular Anaesthesiology and Intensive Care, “Prof. Dr. C.C. Iliescu” Emergency Institute for Cardiovascular Diseases, 258 Fundeni Road, 022328 Bucharest, Romania; 2Faculty of General Medicine, Carol Davila University of Medicine and Pharmacy, 8 Eroii Sanitari Blvd, 050474 Bucharest, Romania; adrian-stefan.dragan2023@stud.umfcd.ro (A.Ş.D.); ovidiu.stiru@umfcd.ro (O.Ş.); 3Department of Cardiovascular Surgery, “Prof. Dr. C.C. Iliescu” Emergency Institute for Cardiovascular Diseases, 258 Fundeni Road, 022328 Bucharest, Romania

**Keywords:** precision medicine, vascular diseases, predictive factors, atherosclerosis, AISI, perioperative risk, aortoiliac occlusive disease

## Abstract

**Objectives:** The role of inflammation in the pathophysiology of atherosclerosis is extensive. Our study aims to assess the predictive role of inflammatory indexes regarding in-hospital mortality in major vascular surgery of Leriche syndrome as a convenient, low-cost, and noninvasive prognostic marker to optimize the patient’s perioperative course. **Methods**: Our retrospective single-center study enrolled consecutive patients diagnosed with aortoiliac occlusive disease, Leriche syndrome, who underwent elective major vascular surgery between 2017 and 2023 in a tertiary cardiovascular center. Preoperative, postoperative, and day-one after-surgery data, including systemic immune-inflammation index (SII), systemic inflammation response index (SIRI), aggregate index of systemic inflammation (AISI), neutrophil–lymphocyte ratio (NLR), platelet–lymphocyte ratio, and monocyte–lymphocyte ratio, were studied to the endpoint, in-hospital death. We also tested the delta values of the indexes to the endpoint. The indexes were compared to the Revised Cardiac Risk Index (RCRI) and Vascular Surgery Group Cardiac Risk Index (VSG-CRI) for outcome prediction. **Results:** The tested inflammatory indexes significantly increased from the preoperative to postoperative and, further, to the day-one settings. Preoperative AISI (*p* = 0.040) emerged as the only independent risk factor regarding in-hospital death occurrence in Leriche patients who underwent major revascularization surgery. While RCRI did not significantly predict the endpoint (AUC = 0.698, *p* = 0.057), VSG-CRI (AUC = 0.864, *p* = 0.001) presented the best result in ROC analysis. Postoperative NLR (AUC = 0.758, *p* = 0.006) was next, followed by NLR postoperative–preoperative (_Preop-_Postop) delta value (AUC = 0.725, *p* = 0.004), postoperative SIRI (AUC = 0.716, *p* = 0.016), SIRI_Preop-_Postop delta value (AUC = 0.712, *p* = 0.016), postoperative SII (AUC = 0.692, *p* = 0.032), and SII_Preop-_Postop delta value (AUC = 0.631, *p* = 0.030). **Conclusions**: Inflammatory indexes are valuable tools for assessing perioperative risk in major vascular surgery, enhancing the value of the already validated risk scores.

## 1. Introduction

Leriche syndrome, also known as aortoiliac occlusive disease, is caused by atherosclerosis affecting the distal abdominal aorta, iliac arteries, and femoropopliteal vessels. Its exact prevalence is unknown as many patients remain asymptomatic for a long time. The therapeutic goals include modifying risk factors, optimizing medical therapies, and performing arterial revascularization of the lower limbs. The most common open surgical alternative for this condition is the aorto-bifemoral/bi-iliac bypass, which involves extensive exposure, aortic clamping, general anesthesia, and sometimes transfusion. This procedure carries high vital risks according to the European Society of Cardiology (ESC) Guidelines [1]. After analyzing data from the Danish Vascular Registry over 20 years, Bredahl et al. reported a 3.6% mortality rate and an overall 20% incidence of major complications within the first 30 days [2]. That is why this type of surgery requires a meticulous preoperative assessment.

The inflammatory burden in Leriche patients who underwent major open surgery is well known. Atherosclerosis is a chronic inflammatory condition that affects elastic and musculo-elastic arteries [3]. Fernández-Friera et al. demonstrated an inflammatory state in the arteries, even in the early stages of atherosclerosis [4]. There is evidence showing the role of inflammatory indexes in assessing the severity of atherosclerosis [5,6,7,8] and their connection to patient outcomes in coronary [9] and carotid disease [10]. Xie et al. reported a positive association between abdominal aortic calcification and the systemic immune-inflammation index (SII) [11], while Zuo et al. found that the monocyte–lymphocyte ratio (MLR) was associated with higher abdominal aortic calcification scores [12]. The recently issued 2024 ESC Guideline for managing peripheral arterial and aortic diseases reveals that, despite heightened emphasis on inflammation in atherosclerotic disease, there are limited reported data regarding peripheral arterial and aortic disease [13].

Additionally, the RELIEF trial recently reported that a significant systemic inflammation after surgery was linked to a higher chance of complications or death during major abdominal surgery [14]. Surgical injury can affect the innate and adaptive immune systems, with the innate immune response occurring first [15]. Changes at the epigenome level, gene expression, cellular differentiation, and effector cell subsets level significantly alter cellular function and activity [15].

There are a variety of biomarkers that can measure the inflammatory status, but they are expensive and sometimes unavailable in some hospitals. That is why there was a great interest in inflammatory assessment using the inflammatory indexes, starting from routine blood analysis.

Our research aims to evaluate the role of inflammatory indexes in predicting in-hospital mortality for major vascular surgery in patients with Leriche syndrome. We are investigating these markers as potential low-cost and noninvasive prognostic indicators. Additionally, we are comparing these indexes to established risk scores in vascular surgery, such as the Revised Cardiac Risk Index (RCRI) [1] and the Vascular Surgery Group Cardiac Risk Index (VSG-CRI) [16]. This analysis will help us determine if these markers can provide valuable information for optimizing medical treatments, wound care, and the timing of surgery on a personalized basis.

## 2. Materials and Methods

Our retrospective single-center study enrolled all the consecutive patients diagnosed with aortoiliac occlusive disease TASC II D who underwent elective major vascular surgery between 2017 and 2023 in “Prof. C. C. Iliescu” Emergency Institute for Cardiovascular Diseases, Bucharest, Romania. The patients with incomplete data were excluded. The study was conducted following the Declaration of Helsinki and approved by the Ethics and Studies Approval Committee of “Prof. C.C. Iliescu” Emergency Institute for Cardiovascular Diseases, Bucharest (No. 14298/21 May 2024).

A vascular surgeon, a cardiologist, and an anesthetist were involved in the multidisciplinary preoperative assessment. The surgical therapeutic approach, the aorto-bifemoral/iliac grafting, was performed under general anesthesia, which was completed in some cases by epidural anesthesia. The patients underwent noninvasive and invasive monitoring. The arterial pressure was measured via an arterial catheter. A central venous catheter was used to monitor the central venous pressure and administer vasopressors. The patients were further cared for in the intensive care unit (ICU) until the resumption of intestinal transit. The intraoperative mechanical ventilation was continued in the ICU until all extubation criteria were met, including respiratory, cardiovascular, neurological, and normothermia parameters.

Our study aims to find the risk factors of in-hospital mortality and the importance of the inflammatory indexes in this setting. The predefined study endpoint was in-hospital death, defined by early postoperative death that occurs during the same hospitalization. The perioperative trend of the inflammatory indexes was also analyzed. We also compared the in-hospital death prediction capacity of the inflammatory indexes (absolute and delta values) to the risk scores (VSC-CRI and RCRI).

The medical preoperative and postoperative data were collected and reviewed in the whole study population and in the two subgroups of patients: non-survivors and survivors. The risk scores, VSC-CRI and RCRI, were used in our analysis as a surrogate for comorbidities. The Leriche–Fontaine Classification stratified the patients according to trophic lesions presence. The studied variables were related to biological sex, age, preoperative creatinine and creatinine clearance (Cockcroft–Gault Equation), and hematological data. We studied SII, SIRI, AISI, NLR, PLR, and MLR as inflammatory indexes preoperatively (_Preop), immediately postoperatively (_Postop), and on day one (_Day1) after surgery to assess our endpoint, in-hospital mortality. The delta values of the inflammatory indexes (_Postop-_Preop, Day1-_Preop, and Day1-_Postop) were also studied in all groups and in the two subgroups of patients. Additionally, the count of the leukocytes (L), neutrophils (N), lymphocytes (Lf), monocytes (M), platelets (P), and the mean platelet volume (MPV) were tested to the same endpoint. There were also collected data about the length of the intraoperative mechanical ventilation, acting as a surrogate for surgery duration, the need for intraoperative bypass graft extension, postoperative surgical reintervention, the duration of postoperative mechanical ventilation, and ICU length of stay. We retrospectively calculated the inflammatory indexes using the following mathematical formulas:
 SII = N × P/Lf; SIRI = N × M/Lf; AISI = N × PxM/Lf; NLR =
N/Lf; PLR = P/Lf; MLR = M/Lf.

For statistical analysis, SPSS 29 was used to statistically analyze the data, with a threshold of statistical significance of 95% (*p* ≤ 0.05). After checking for normality of distribution (the Shapiro–Wilk test), the quantitative variables were displayed using mean ± standard deviation or mean and interquartile range (IQR) and further evaluated in the two subgroups using an independent *t*-test or Mann–Whitney test. Categorical variables were assessed using the Fisher test. The inflammatory indexes were graphically presented using a box plot to illustrate their trend. To assess the statistical significance of the change in the inflammatory index values related to _Day1, _Postop, and _Preop settings, we used the nonparametric test, the Friedman test (χ(2), *p*), as the data were not normally distributed. The Wilcoxon signed test (Z, *p*) with Bonferroni correction (*p* = 0.05/3 = 0.016) was chosen as the post hoc test. Each variable was tested separately in univariable binary logistic regression. After testing for multicollinearity, the variables with variance inflation factors lower than five were introduced in the multivariable binary logistic regression to identify the independent risk factors (OR, CI95%, *p*) for in-hospital mortality (Enter analysis and Backward analysis when Enter analysis failed to find a significant result). We used the receiver operator characteristic (ROC) analysis to assess the capacity of the quantitative variables (absolute and delta values of the inflammatory indexes, RCRI, and VSC-CRI) to predict the endpoint (AUC, *p*, CI 95%) and to compare them in this setting. The cut-offs (based on the Youden index) with their sensitivity and specificity were displayed if the AUC was statistically significant.

## 3. Results

### 3.1. Data Presentation in the Study Population and in the Two Subgroups: Survivors and Non-Survivors

We reviewed 116 consecutive patients previously diagnosed with aortoiliac occlusive disease TASC II D who underwent major vascular surgery. Aorto-bifemoral/iliac grafting was performed in the “Prof. C. C. Iliescu” Emergency Institute for Cardiovascular Diseases, Bucharest, Romania, between 2017 and 2023. This retrospective study enrolled 114 consecutive patients, as 2 patients with incomplete data were excluded from the study. Five in-hospital deaths (non-survivors) were encountered (5/114, 4.38%). A diagram of our study is presented in Figure 1.

According to the Leriche–Fontaine Classification, 70 patients (61.40%) were diagnosed as stage III, while 44 patients (38.59%) with trophic lesions (necrosis or gangrene) were classified as stage IV of the disease. The patients, aged 60.34 (±7.12) years old, mostly men (103/114, 90.35%), underwent surgery under general anesthesia. The intraoperative mechanical ventilation was 7 (6–8) hours. Intraoperatively, epidural anesthesia was added to general anesthesia in 68 (59.64%) patients. The epidural anesthesia did not significantly determine the in-hospital mortality (*p* = 1). Some patients (34/114, 29.82%) required an extension of the graft to the popliteal artery. After the procedure, patients were placed on mechanical ventilation in the ICU until they met all the necessary extubation criteria. Others (19/114, 16.67%) underwent surgical reintervention. Table 1 presents the demographic characteristics and the studied variables in the study population and the two groups under study: survivors and non-survivors.

We used RCRI and VSG-CRI as preoperative risk scores to assess the cardiovascular and mortality risk (Table 2). The studied patients presented an RCRI of 2 (1–3) and a VSG-CRI of 4 (3–6). We found that 80% of the in-hospital death occurred in patients with VSG-CRI ≥7, while all non-survivors presented RCRI ≥2. Survivors presented an RCRI of 2 (1–3) and a VSG-CRI of 4 (2.5–6). Non-survivors had an RCRI of 2 (2–4) and a VSG-CRI of 7 (6–8.5). Only VSG-CRI had significantly higher values in non-survivors compared to survivors (*p* = 0.005).

The mean age of non-survivors was 68.80 (±4.02) years old, significantly higher than the mean age of 59.95 (±7.00) years old in the survivors’ group (*p* = 0.005). Non-survivors experienced longer lengths of stay and mechanical ventilation in the ICU when compared to survivors (*p* = 0.001 and *p* = 0.030, respectively, Table 1). The intraoperative mechanical ventilation, a surrogate for the duration of the surgery, and the type of anesthesia were not significantly different in the two groups (Table 1). Surgical reintervention was significantly associated with in-hospital death (*p* = 0.003). Four of the five non-survivors (80%) had a surgical reintervention.

The total leukocytes (preoperative and day-1 level) and platelets (preoperative, postoperative, and day-1 level) count was higher in non-survivors, although the result was not statistically significant (Table 1). Regarding the leukocyte formula, except for the preoperative and postoperative level of the lymphocytes, all the other components presented higher levels in the non-survivor group but without significance (Table 1). The survivor group had a significantly lower level of postoperative leukocytes (median 11.05, IQR 8.43–14.19) and postoperative neutrophils (median 8.54, IQR 10.43–15.49) by comparison with the non-survivors group.

Non-survivors exhibited higher inflammatory indexes when compared to survivors, but the results were not statistically significant, as presented in Table 1. The difference of the inflammatory values (delta) regarding the three perioperative moments (_Postop-_Preop, _Day1- _Preop, and _Day1- _Postop) were not significantly different when comparing the two subgroups of patients (Table 1). However, in the studied population, the tested inflammatory indexes significantly increased from the preoperative to postoperative and, further, to day 1 settings (*p* = 0.001 in Friedman analysis, Table 3). The post hoc analysis demonstrated statistically significant results between the groups (_Postop-_Preop, _Day1- _Preop, and _Day1- _Postop) (*p* < 0.001, Table 3), except for the PLR_Postop-PLR_Preop (*p* = 0.021, result without statistical significance in Bonferroni correction). We also studied the trend of the inflammatory indexes in the studied three perioperative moments in the two subgroups of patients. We found in the survivors group the same results as in the whole study population (Table 3). In contrast, in the non-survivors group, SII values were not significantly different between the three perioperative moments (*p* = 0.091). The post hoc analysis did not succeed in finding significant results when comparing the other indexes in the three perioperative moments (Table 3). Figure 2 boxplots show the trend of the inflammatory indexes related to preoperative, postoperative, and day one moments in the two subgroups of patients.

### 3.2. The In-Hospital Mortality Risk Factors Analysis

The variables were tested one by one in the logistic binary regression in univariable analysis to the endpoint (in-hospital death). The total leukocytes (preoperative and day 1 count), neutrophils (preoperative, postoperative, and day 1 count), and monocytes (preoperative, postoperative, and day 1 count) presented significant results (*p* ≤ 0.05) (Table 4). The preoperative AISI (*p* = 0.033), NLR (*p* = 0.034), and MLR (*p* = 0.050) were risk factors for in-hospital death in univariable analysis (Table 4). The immediately postoperative SIRI (*p* = 0.033) and MLR (*p* = 0.038) presented the same result (Table 4). From day 1 inflammatory indexes, only SIRI (*p* = 0.038) and the difference between day 1 and preoperative SIRI value (*p*= 0.0036) had significant results in logistic binary regression in univariable analysis. The age, the biological sex of the patients, and the VSC-CRI score, as well as the presence of the surgical reintervention, were also risk factors in univariable analysis (*p* ≤ 0.05).

After testing for multicollinearity, the variables with variance inflation factors higher than five were excluded. The model, including biological sex, surgical reintervention, age, preoperative AISI, and VSC-CRI, was tested in the multivariable binary logistic regression. The biological sex, age, and VSC-CRI did not present significant results in multivariable binary logistic regression (*p* = 0.356, *p* = 0.083, and *p* = 0.334, respectively), while the surgical reintervention result was near the significance level (*p* = 0.051). We report a statistically significant result in multivariable binary logistic regression regarding the tested endpoint, in-hospital death, only for the preoperative AISI (*p* = 0.040, Enter analysis) (Table 4). Preoperative AISI (OR 1.001, CI 95% 1–1.002) was the only independent risk factor of in-hospital death (Table 4).

### 3.3. The ROC Analysis

ROC analysis was used to classify the variables predicting in-hospital death endpoint in the studied population (Table 5).

VSG-CRI presented the highest AUC with statistical significance (AUC 0.864, *p* = 0.001, CI95% 0.748–0.981), with a cut off of 6.5 (sensitivity 80%; specificity 85.3%). In contrast, AUC of RCRI was not statistically significant (*p* = 0.057) (Figure 3). None of the preoperatory or Day 1 inflammatory indexes had statistically significant results in the ROC analysis when predicting in-hospital death (Table 5). Only immediately postoperative NLR (AUC 0.758, *p* = 0.006, CI95%: 0.575–0.941), the NLR_Postop-NLR_Preop delta value (AUC 0.725, *p* = 0.004, CI 95%: 0.571–0.878), SIRI_Postop (AUC 0.716, *p* = 0.016, CI95%: 0.540–0.891), the SIRI_Postop-SIRI_Preop delta value (AUC 0.712, *p* = 0.020, CI 95%: 0.534–0.890), SII_Postop (AUC 0.692, *p* = 0.032, CI95%: 0.517–0.867), and the SII_Postop-SII_Preop delta value (AUC 0.631, *p* = 0.030, CI 95%: 0.512–0.750) presented good significant results, predicting in-hospital death better than RCRI but less accurately compared to VSG-CRI (Figure 4).

## 4. Discussion

We reported the role of the inflammatory indexes in major open surgery of Leriche syndrome when targeting in-hospital death. The tested inflammatory indexes increased from the preoperative to postoperative and, further, to day 1 settings, with higher values in the non-survivors group. Preoperative AISI was the independent risk factor concerning in-hospital death. When comparing the inflammatory indexes to the risk scores, immediately postoperative NLR, SII, and SI and the postoperative–preoperative delta value of the NLR, SII, and SII were superior to the RCRI but not to VSG-CRI in this setting.

The innate immune response occurs in surgical trauma [15]. Monocytes and neutrophils are activated to produce proinflammatory mediators [15,17]. Additionally, monocytes interacting with platelets can promote inflammatory and prothrombotic pathways [17,18,19]. Platelets can also interact with neutrophils, enhancing the systemic inflammatory response [17,18]. On the other hand, lymphocytes play an essential role in the anti-inflammatory response [17].

AISI, the aggregate index of systemic inflammation, which is the least studied marker of its kind in the medical literature, represents a complex index containing information regarding neutrophils, monocytes, platelets, and lymphocytes, information that is easy and cheap to obtain. We demonstrated its role in major open surgery of Leriche patients. Studying the preoperative AISI in this setting may help identify patients who could benefit from a personalized preoperative medical approach, optimizing the timing of the surgery. The wound healing strategy may need to be adjusted for these patients, with negative-pressure therapy showing promising results in this context [20]. Medical therapy should also be personalized for these patients. The use of angiotensin-converting enzyme inhibitors for antihypertensive treatment has been shown to have anti-inflammatory effects [21]. Statins also play a role in this scenario. The JUPITER trial demonstrated a reduction in cardiovascular events with the use of rosuvastatin in patients with low cholesterol levels but high systemic inflammation [22]. Canakinumab Anti-inflammatory Thrombosis Outcomes Study (CANTOS) showed that IL-1β inhibition by a monoclonal antibody substantially reduced cardiovascular disease risk [22]. Optimizing the treatment of diabetes mellitus, microbiome, and nutrition status might be also required [21]. Colchicine with pleiotropic anti-inflammatory effects in cardiovascular diseases [23] did not present significant results in vascular patients yet [24].

When comparing the inflammatory indexes with the risk scores, we demonstrated that immediately postoperative NLR, SII, and SIRI predicted more accurately in-hospital death compared to RCRI but not to VSG-CRI. Previously, Bertges et al. reported that RCRI underestimated in-hospital cardiac events in patients undergoing elective or urgent vascular surgery, while the VSG-CRI more accurately predicted in-hospital cardiac events after vascular surgery [16]. Moreover, the intraoperative course can determine the postoperative inflammatory pattern. Surgery duration, surgical approach (retroperitoneal/transperitoneal), intraoperative transfusion, hypotension, and anesthetic management can modulate the postoperative inflammatory response and further complications and even death [15,25,26,27,28].

In vascular surgery, the inflammatory indexes were studied, especially NLR, SII, and SIRI. Pasqui et al. demonstrated that NLR, PLR, and SII predicted the occurrence of delirium [29]. High preoperative MLR, NLR, PLR, SII, SIRI, and AISI were independent factors predicting the risk of 12-month restenosis and mortality following carotid endarterectomy in the analysis of Niculescu et al. [30]. Erturk et al. found that an increased NLR was associated with higher cardiovascular mortality in patients with peripheral arterial occlusive disease admitted with critical limb ischemia or intermittent claudication [31]. Preoperative NLR and PLR predicted amputation in patients with peripheral artery disease or those who underwent a revascularization intervention [32,33]. A high NLR value was an independent predictor of mortality in endovascular aneurysm repair [34]. Coelho et al. also reported that higher preoperative NLR was associated with 30-day death or amputation in acute limb ischemia revascularization [35]. NLR was an independent risk factor of mortality in the study by Adler et al. regarding patients undergoing open lower extremity revascularizations [36].

The inflammatory indexes were tested in other settings. SIRI, SII, AISI, and NLR were associated with in-hospital mortality in aortic valve replacement [37]. SIRI was the strongest predictor of a poor outcome in aortic valvular surgery [37]. High preoperative SII values predicted prolonged mechanical ventilation and ICU stay after off-pump coronary artery bypass grafting surgery [38] and mortality [39]. Urbanowicz et al. reported NLR and SIRI as predictors of mortality in off-pump coronary artery bypass surgery [40]. Zhu et al. found that elevated NLR was associated with higher mortality and ICU admission across many procedures [41]. AISI helped identify patients at risk of prolonged hospital stays in open elective thoracic surgery [42]. AISI predicted the prognosis of stroke [43] and hypertension [44] and was an independent risk factor for in-stent restenosis following drug-eluting stent implantation [45]. SII, SIRI, and AISI improved the accuracy of predicting the incidence of major adverse cardiac events in patients with myocardial infarction with nonobstructive coronary arteries [46].

The study has limitations due to its retrospective and single-center nature and the discrepancy between the number of survivors and non-survivors, which restrict the data and subsequent analysis. To our knowledge, this paper is the first to assess the role of the inflammatory indexes in a Leriche major open revascularization setting. Prospective research is required. Additionally, studies are needed to identify other cost-effective elements from routine blood analysis that can aid in risk-stratifying Leriche patients. Furthermore, research might evaluate the relationship between inflammatory markers and preoperative medication, preoperative wound care methods, and intraoperative anesthetic approaches.

## 5. Conclusions

The inflammatory indexes might be useful tools in assessing the perioperative risk in major vascular surgery. Preoperative AISI emerged as the only independent risk factor regarding in-hospital death occurrence in Leriche patients who undergo major revascularization surgery. Postoperative NLR, SII, and SI were superior to RCRI but not to VSG-CRI in predicting in-hospital death.

## Figures and Tables

**Figure 1 jcm-13-06338-f001:**
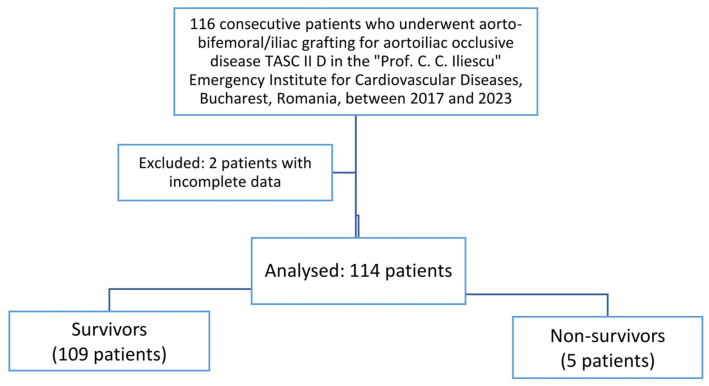
The diagram of the study.

**Figure 2 jcm-13-06338-f002:**
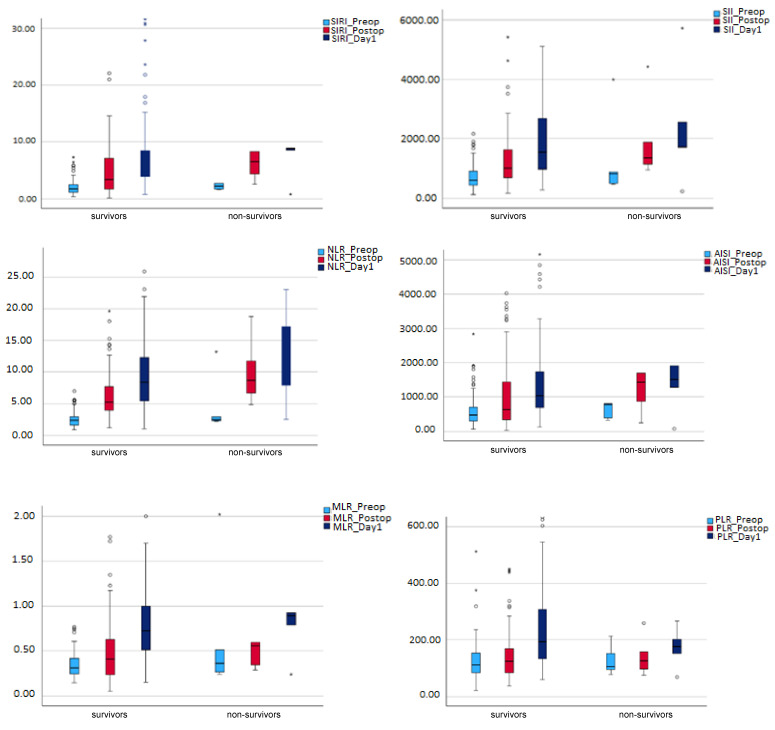
The trend of the tested inflammatory indexes from the preoperative (_Preop) to postoperative (_Postop) and day 1 settings (_Day1) in the two subgroups: survivors and non-survivors. Abbreviations: AISI, aggregate index of systemic inflammation; _Day1, day one value; MLR, monocyte-to-lymphocyte ratio; NLR, neutrophil-to-lymphocyte ratio; PLR, platelet-to-lymphocyte ratio; _Postop, postoperative; _Preop, preoperative; SII, systemic inflammatory index; SIRI, systemic inflammatory response index.

**Figure 3 jcm-13-06338-f003:**
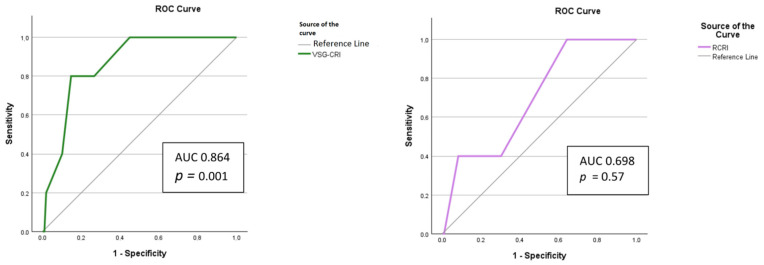
The receiver operator characteristic curve of VSG-CRI and RCRI predicting in-hospital death. Abbreviations: AUC, area under the curve; *p* probability value; RCRI, Revised Cardiac Risk Index; ROC, receiver operator characteristic; VSC-CRI, Vascular Surgery Group Cardiac Risk Index.

**Figure 4 jcm-13-06338-f004:**
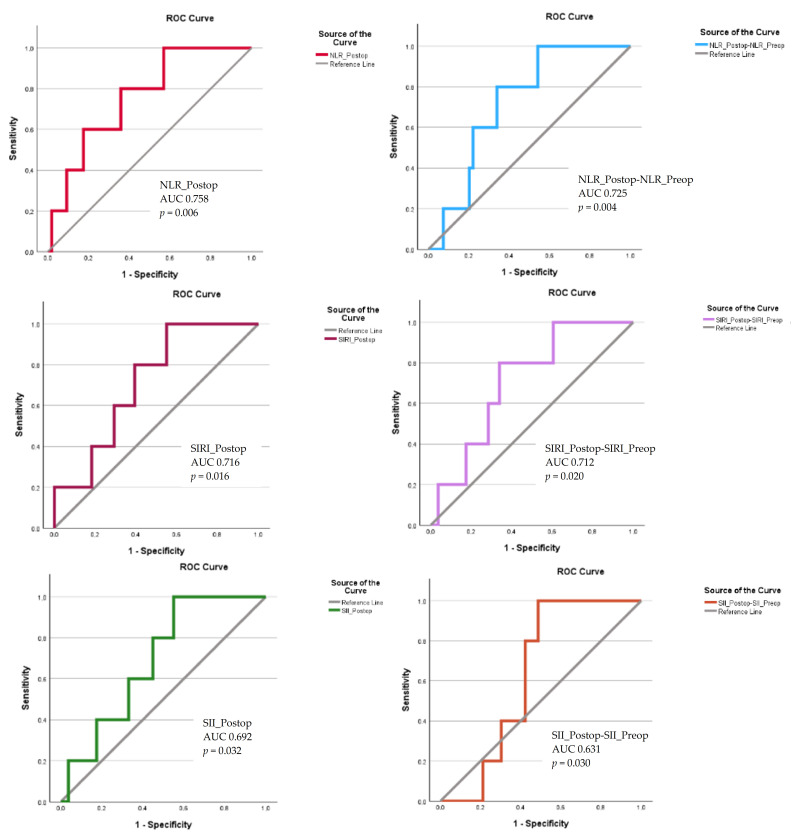
The ROC curve of absolute or delta values of the inflammatory indexes predicting in-hospital death with a statistically significant result. Abbreviations: AUC, area under the curve; NLR_Postop, postoperative neutrophil-to-lymphocyte ratio; NLR_Preop, preoperative neutrophil-to-lymphocyte ratio *p* probability value; ROC, receiver operator characteristic; SII_Postop, postoperative systemic inflammatory index; SII_Preop, preoperative systemic inflammatory index; SIRI_Postop, postoperative systemic inflammatory response index; SIRI_Preop, preoperative systemic inflammatory response index.

**Table 1 jcm-13-06338-t001:** Data studied in the study population and the two subgroups: survivors and non-survivors.

Variable		Leriche Patients (*n* = 114)	Non-Survivors (*n* = 5)	Survivors (*n* = 109)	*p*
Age (years)		60.34 (±7.12)	68.80 (±4.02)	59.95 (±7.00)	0.005
RCRI		2 (1–3)	2 (2–4)	2 (1–3)	0.117
VSG-CRI		4 (3–6)	7 (6–8.5)	4 (2.5–6)	0.005
Sex	male	103 (90.35%)	3 (2.91%)	100 (97.08%)	0.072
	female	11 (9.64%)	2 (18.18%)	9 (81.81%)	
Leriche-Fontaine	III	70 (61.40%)	3 (4.28%)	67 (95.72%)	1.00
	IV	44 (38.59%)	2 (4.54%)	42 (95.46%)	
Creat_Preop (mg/dL)		0.84 (0.72–1)	0.88 (±0.25)	0.84 (0.72–1)	0.994
Clear_Creat_Preop (ml/min)		98 (84.75–105)	83.60 (±23.92)	99 (85–105)	0.262
Leu_Preop (×10^3^/μL)		9.19 (7.80–10.91)	12.69 (±6.38)	9.46 (± 2.39)	0.320
N_Preop (×10^3^/μL)		5.54 (4.48–6.85)	8.80 (±1.96)	5.78 (4.47–6.83)	0.189
Lf_Preop (×10^3^/μL)		2.40 (1.90–2.97)	2.36 (±0.90)	2.42 (1.92–2.98)	0.609
P_Preop (×10^3^/μL)		276 (211.75–339.25)	276 (±61.95)	275 (211.50–339.50)	0.967
M_Preop (×10^3^/μL)		0.77 (0.61–0.95)	0.91 (0.74–1.91)	0.76 (0.61–0.95)	0.175
MPV_Preop (fl)		10.40 (1.15)	11.02 (±0.73)	10.37 (±1.16)	0.125
SII_Preop		618.29 (437.99–906.48)	814.14 (484.58–2438.51)	608.34 (430.45–906.52)	0.395
SIRI_Preop		1.75 (1.20–2.49)	2.21 (1.65–20.29)	1.75 (1.13–2.48)	0.182
AISI_Preop		472.10 (307.22–742.85)	773.43 (359.21–6133.63)	468.73 (298.72–721.79)	0.271
NLR_Preop		2.34 (1.59–2.94)	2.43 (2.23–8.04)	2.33 (1.56–2.96)	0.284
MLR_Preop		0.31 (0.24–0.42)	0.36 (0.25–1.26)	0.31 (0.24–0.41)	0.387
PLR_Preop		112.10 (85.03–153.74)	129.05 (±54.36)	112.12 (84.98–153.80)	0.798
Bypass extension	absent	80 (70.17%)	3 (3.75%)	77 (96.25%)	0.634
	present	34 (29.82%)	2 (5.88%)	32 (94.11%)	
Anesthesia	GA + E	68 (59.64%)	3 (4.41%)	65 (95.58%)	1
	GA	46 (40.35%)	2 (4.34%)	44 (95.65%)	
Leu_Postop (×10^3^/μL)		11.10 (8.51–14.40)	15.73 (±3.96)	11.05 (8.43–14.19)	0.035
N_Postop (×10^3^/μL)		8.73 (6.68–11.80)	12.92 (±2.92)	8.54 (10.43–15.49)	0.015
Lf_Postop (×10^3^/μL)		1.58 (1.16–2.34)	1.55 (±0.79)	1.60 (0.84–2.36)	0.585
P_Postop (×10^3^/μL)		198.50 (160–246.25)	190.40 (±54.23)	198 (147–226.50)	0.950
M_Postop (×10^3^/μL)		0.62 (0.37–1.02)	1.16 (±0.92)	0.61 (0.49–1.96)	0.248
MPV_Postop (fl)		10.35 (9.47–10.90)	11.00 (±0.51)	10.23 (±1.05)	0.125
SII_Postop		1028.42 (714.67–1658.32)	1360.80 (1046.92–3146.48)	1013. 10 (677.84–1636.89)	0.148
SIRI_Postop		3.44 (1.72–7.24)	6.56 (3.49–29.14)	3.38 (3.49–29.14)	0.104
AISI_Postop		635.37 (343.19–1483.84)	1424.09 (559.81–6750.09)	630.44 (343.03–1449.07)	0.226
NLR_Postop		5.29 (3.99–7.84)	10.11 (±5.44)	5.22 (3.91–7.70)	0.052
MLR_Postop		0.41 (0.23–0.62)	0.55 (0.31–1.76)	0.41 (0.23–0.62)	0.316
PLR_Postop		124.29 (83.64–170.24)	143.24 (±72.03)	123.78 (83.18–171.89)	0.874
Leu_Day1 (×10^3^/μL)		10.16 (8.24–12.16)	14.85 (±10.79)	10 (8.24–12.01)	0.303
N_Day1 (×10^3^/μL)		7.95 (6.39–9.71)	12.41 (±9.34)	7.88 (6.32–9.63)	0.191
Lf_Day1 (×10^3^/μL)		1.03 (0.60–1.47)	1.10 (±0.35)	1.03 (0.60–1.48)	0.906
P_Day1 (×10^3^/μL)		200.50 (154.00–249.00)	174.80 (±60.76)	201 (155.50–249.50)	0.340
M_Day1 (×10^3^/μL)		0.73 (0.52–0.96)	0.88 (0.41–2.39)	0.73 (0.53–0.95)	0.688
MPV_Day1 (fl)		10.75 (10.00–11.52)	10.90 (10.80–11.65)	10.70 (9.95–11.55)	0.346
SII_Day1		1575.17 (966.54–2682.73)	2383.20 (±2043.08)	1550.30 (965.82–2688.54)	0.594
SIRI_Day1		5.65 (3.83–8.67)	8.82 (4.69–46.60)	5.42 (3.76–8.55)	0.216
AISI_Day1		1057.34 (679.73–1901.72)	1504.85 (676.03–11443.67)	1043.10 (674.76–1816.46)	0.451
NLR_Day1		8.38 (5.37–12.48)	12.13 (±8.01)	8.35 (5.19–20.06)	0.365
MLR_Day1		0.72 (0.51–1.00)	1.16 (±1.05)	0.72 (0.51–1.95)	0.502
PLR_Day1		191.97 (133.53–302.51)	173.22 (±72.22)	194.17 (110.38–234.25)	0.403
SII_Postop-SII_Preop		436.84 (68–942.10)	486.21 (451.47–849.06)	391.99 (50.97–942.37)	0.322
SII_Day 1-SII_Preop		921.29 (347.36–1862)	888.00 (294.34–1889.50)	922.26 (347.07–1862.25)	0.950
SII_Day 1-SII_Postop		508.44 (−2.08–1408.36)	349.25 (−448.35–1357.92)	551.55 (10.23–1436.10)	0.633
SIRI_Postop-SIRI_Preop		1.61 (0.24–4.68)	3.81 (1.98–9.12)	1.53 (0.14–4.47)	0.110
SIRI_Day 1-SIRI_Preop		3.47 (1.93–6.5)	6.73 (2.64–26.64)	3.39 (1.87–6.28)	0.231
SIRI_Day 1-SIRI_Postop		2.37 (−0.02–4.32)	2.26 (−1.47–20.13)	2.40 (−0.05–4.28)	0.663
AISI_Postop-AISI_Preop		189.94 (−114.96–861.85)	540.90 (97.64–774.27)	182.83 (187.65–1228.38)	0.485
AISI_Day 1_AISI_Preop		541.93 (190.64–1225.13)	888.21 (223.57–5324.87)	532.04 (187.65–1228.38)	0.502
AISI_Day 1-AISI_Postop		468.24 (−1.11–877.24)	482.30 (−495.82–5104.85)	467.43 (10.34–851.09)	0.852
NLR_Postop-NLR_Preop		2.94 (1.52–5.28)	5.54 (3.38–7.64)	2.87 (1.44–5.07)	0.090
NLR_Day 1-NLR_Preop		5.72 (3.1–9.7)	7.74 (3.10–9.66)	5.70 (3.10–9.66)	0.556
NLR_Day 1-NLR_Postop		2.63 (−0.13–5.68)	3.54 (−1.55–4.82)	2.81 (−0.05–6.18)	0.663
MLR_Postop-MLR_Preop		0.05 (−0.05–0.26)	0.07 (−0.01–0.63)	0.05 (−0.05–0.25)	0.476
MLR_Day 1-MLR_Preop		0.38 (0.21–0.67)	0.53 (0.12–0.82)	0.38 (0.20–0.66)	0.787
MLR_Day 1-MLR_Postop		0.32 (0.08–0.53)	0.23 (−0.02–0.46)	0.33 (0.08–0.53)	0.426
PLR_Postop-PLR_Preop		4.02 (−19.52–41.7)	6.50 (−0.95–33.18)	3.85 (−21.11–47.56)	0.704
PLR_Day 1-PLR_Preop		84.06 (32.06–160.42)	0.22 (−10.60–120.92])	85.04 (33.73–162.91)	0.141
PLR_Day 1-PLR_Postop		69.15 (18.4–151.39)	−6.24 (−31.59–109.66)	69.44 (21.22–154.75)	0.148
ICU_hours		68 (53.75–90)	465.40 (±476.85)	67 (49–90)	0.001
MV_ICU (hours)		8 (5–14)	15 (9.5–861)	8 (5–13.50)	0.030
MV_intraop (hours)		7 (6–8)	8.8 (±2.95)	7 (6–8)	0.090
Surgical reintervention	absent	95 (83.33%)	1 (1.09%)	90 (98.90%)	0.003
	present	19 (16.67%)	4 (21.05%)	15 (78.94%)	

Abbreviations: AISI_Day1, aggregate index of systemic inflammation from day one after surgery; AISI_Preop, preoperative aggregate index of systemic inflammation; AISI_Postop, postoperative aggregate index of systemic inflammation; Clear_Creat_Preop, preoperative creatinine clearence; Creat_Preop, preoperative creatinine; E, epidural anesthesia; GA, general anesthesia; ICU, intensive care unit; Leu_Day1, leukocytes count from day one after surgery; Leu_Preop, preoperative leukocytes count; Leu_Postop, postoperative leukocytes count; Lf_Day1, lymphocytes from day one after surgery; Lf_Preop, preoperative lymphocytes; Lf_Postop, postoperative lymphocytes; M_Day1, monocytes count from day one after surgery; M_Postop, postoperative monocytes count; M_Preop, preoperative monocytes count; MLR_Day1, monocyte-to-lymphocyte ratio from day one after surgery; MLR_Postop, postoperative monocyte-to-lymphocyte ratio; MLR_Preop, preoperative monocyte-to-lymphocyte ratio; MPV_Day1, mean platelet volume from day one after surgery; MPV_Preop, preoperative mean platelet volume, MPV_Postop, postoperative mean platelet volume; MV_intraop, intraoperative mechanical ventilation; MV_ICU, mechanical ventilation in intensive care unit; N_Day1, neutrophils count from day one after surgery; N_Preop, preoperative neutrophils count; N_Postop, postoperative neutrophils count; NLR_Day1, neutrophil-to-lymphocyte ratio from day one after surgery; NLR_Preop, preoperative neutrophil-to-lymphocyte ratio; NLR_Postop, postoperative neutrophil-to-lymphocyte ratio; *p* probability value; P_Day1, platelet count from day one after surgery; P_Preop, preoperative platelet count; P_Postop, postoperative platelet count; PLR_Day1, platelet-to-lymphocyte ratio from day one after surgery; PLR_Preop, preoperative platelet-to-lymphocyte ratio; PLR_Postop, postoperative platelet-to-lymphocyte ratio; SII_Day1, systemic inflammatory index from day one after surgery; SII_Preop, preoperative systemic inflammatory index; SII_Postop, postoperative systemic inflammatory index; SIRI_Day1, systemic inflammatory response index from day one after surgery; SIRI_Preop, systemic inflammatory response index; SIRI_Postop, postoperative systemic inflammatory response index.

**Table 2 jcm-13-06338-t002:** RCRI and VSG-CRI scores in our cohort.

Risk Score	Value	No. Patients (*n*, %)	Non-Survivors (*n* = 5)
VSG-CRI	0–3	35 (30.70%)	0
	4–6	59 (51.75%)	1
	≥7	20 (17.54%)	4
RCRI	1	39 (34.21%)	0
	2	40 (35.08%)	3
	3	24 (21.05%)	0
	≥4	11 (9.64%)	2

Abbreviations: No, number; RCRI, Revised Cardiac Risk Index; VSC-CRI, Vascular Surgery Group Cardiac Risk Index.

**Table 3 jcm-13-06338-t003:** The statistical analysis of the trend of the inflammatory indexes related to preoperative, postoperative, and day one setting in the whole group and in the subgroups with post hoc tests.

Var	Friedman Analysis						Wilcoxon Signed Test with Bonferroni Correction (Corrected *p* = 0.016)						
	All Group		Survivors		Non-Survivors		Combinations	All Group		Survivors		Non-Survivors	
	χ(2)	*p*	χ(2)	*p*	χ(2)	*p*		Z	*p*	Z	*p*	Z	*p*
SII	119.68	0.001	115.92	0.001	4.8	0.091	SII_Postop-SII_Preop	−7.56	<0.001	−7.27	<0.001	-	-
							SII_Day 1-SII_Preop	−8.82	<0.001	−8.66	<0.001	-	-
							SII_Day 1-SII_Postop	−5.25	<0.001	−5.18	<0.001	-	-
SIRI	123.59	0.001	115.92	0.001	8.4	0.015	SIRI_Postop-SIRI_Preop	−7.17	<0.001	−6.93	<0.001	−1.75	0.080
							SIRI_Day 1-SIRI_Preop	−9.11	<0.001	−8.89	<0.001	−2.02	0.043
							SIRI_Day 1-SIRI_Postop	−4.68	<0.001	−4.27	<0.001	−2.02	0.043
AISI	84.84	0.001	77.43	0.001	8.4	0.015	AISI_Postop-AISI_Preop	−4.64	<0.001	−4.39	<0.001	−1.75	0.080
							AISI_Day 1- AISI_Preop	−8.38	<0.001	−8.12	<0.001	−2.02	0.043
							AISI_Day 1-AISI_Postop	−4.46	<0.001	−4.09	<0.001	−2.02	0.043
NLR	168.09	0.001	158.76	0.001	10	0.007	AISI_Postop-AISI_Preop	−9.02	<0.001	−8.81	<0.001	−2.02	0.043
							AISI_Day 1-AISI_Preop	−9.24	<0.001	−9.03	<0.001	−2.02	0.043
							AISI_Day 1-AISI_Postop	−5.58	<0.001	−5.20	<0.001	−2.02	0.043
PLR	86.01	0.001	79.09	0.001	7.6	0.022	PLR_Postop-PLR_Preop	−2.30	0.021	−2.30	0.021	−0.13	0.893
							PLR_Day 1-PLR_Preop	−8.30	<0.001	−8.05	<0.001	−2.02	0.043
							PLR_Day 1-PLR_Postop	−7.19	<0.001	−6.94	<0.001	−2.02	0.043
MLR	133.56	0.001	123.86	0.001	8.4	0.015	MLR_Postop-MLR_Preop	−4.38	<0.001	−4.14	<0.001	−1.21	0.225
							MLR_Day 1-MLR_Preop	−9.10	<0.001	−8.85	<0.001	−2.02	0.043
							MLR_Day1-MLR_Postop	−7.18	<0.001	−6.85	<0.001	−2.02	0.043

The results are presented as χ(2), *p* in Friedman analysis and Z, *p* in Wilcoxon signed test with Bonferroni correction (corrected *p* = 0.016), used as post hoc test. Abbreviations: AISI_Day1, aggregate index of systemic inflammation from day one after surgery; AISI_Preop, preoperative aggregate index of systemic inflammation; AISI_Postop, postoperative aggregate index of systemic inflammation; MLR_Day1, monocyte-to-lymphocyte ratio from day one after surgery; MLR_Postop, postoperative monocyte-to-lymphocyte ratio; MLR_Preop, preoperative monocyte-to-lymphocyte ratio; NLR_Day1, neutrophil-to-lymphocyte ratio from day one after surgery; NLR_Preop, preoperative neutrophil-to-lymphocyte ratio; NLR_Postop, postoperative neutrophil-to-lymphocyte ratio; *p* probability value; PLR_Day1, platelet-to-lymphocyte ratio from day one after surgery; PLR_Preop, preoperative platelet-to-lymphocyte ratio; PLR_Postop, postoperative platelet-to-lymphocyte ratio; SII_Day1, systemic inflammatory index from day one after surgery; SII_Preop, preoperative systemic inflammatory index; SII_Postop, postoperative systemic inflammatory index; SIRI_Day1, systemic inflammatory response index from day one after surgery; SIRI_Preop, systemic inflammatory response index; SIRI_Postop, postoperative systemic inflammatory response index; PLR_Day1, platelet-to-lymphocyte ratio from day one after surgery; PLR_Preop, preoperative platelet-to-lymphocyte ratio; PLR_Postop, postoperative platelet-to-lymphocyte ratio; RCRI, Revised Cardiac Risk Index; SII_Day1, systemic inflammatory index from day one after surgery; SII_Preop, preoperative systemic inflammatory index; SII_Postop, postoperative systemic inflammatory index; SIRI_Day1, systemic inflammatory response index from day one after surgery; SIRI_Preop, systemic inflammatory response index; SIRI_Postop, postoperative systemic inflammatory response index; var, variable.

**Table 4 jcm-13-06338-t004:** The binary logistic regression results (endpoint: in-hospital death) in the study population (*n* = 114).

Variable	Univariable Binary Logistic Regression		Multivariable Binary Logistic Regression	
	OR (CI 95%)	*p*	OR (CI 95%)	*p*
Leu_Preop	1.311 (1.034–1.683)	0.026		
N_Preop	1.353 (1.048–1.747)	0.020		
Lf_Preop	-	0.621		
P_Preop	-	0.799		
M_Preop	7.145 (1.216–41.972)	0.029		
MPV_Preop	-	0.222		
SII_Preop	-	0.360		
SIRI_Preop	-	0.183		
AISI_Preop	1.001 (1–1.002)	0.033	1.001 (1–1.002)	0.040
NLR_Preop	1.478 (1.030–2.121)	0.034		
MLR_Preop	21.513 (0.998–463.676)	0.050		
PLR_Preop	-	0.989		
Leu_Postop	-	0.063		
N_Postop	1.216 (1.008–1.467)	0.041		
Lf_Postop	-	0.593		
P_Postop	-	0.619		
M_Postop	4.896 (1.007–23.818)	0.049		
MPV_Postop	-	0.115		
SII_Postop	-	0.583		
SIRI_Postop	1.114 (1.009–1.230)	0.033		
AISI_Postop	-	0.164		
NLR_Postop	-	0.102		
MLR_Postop	4.369 (1.086–17.579)	0.038		
PLR_Postop	-	0.848		
Leu_Day1	1.166 (1.007–1.351)	0.040		
N_Day1	1.194 (1.011–1.411)	0.037		
Lf_Day1	-	0.882		
P_Day1	-	0.293		
M_Day1	3.460 (1.021–11.727)	0.046		
MPV_Day1	-	0.510		
SII_Day1	-	0.781		
SIRI_Day1	1.064 (1.003–1.127)	0.038		
AISI_Day1	-	0.088		
NLR_Day1	-	0.233		
MLR_Day1	-	0.136		
PLR_Day1	-	0.352		
Age	1.280 (1.062–1.542)	0.010	1.463 (0.951–2.249)	0.083
Surgical reintervention	24.00 (2.508–229.638)	0.006	583 (0.977–34,835.419)	0.051
MV_intraop	-	0.061		
Anesthesia	-	0.967		
Bypass extension	-	0.614		
Sex	7.407 (1.092–50.264)	0.040	19.846 (0.035–11,241.119)	0.579
Trophyc lesions	-	0.947		
Clear_Creat_Preop	-	0.294		
RCRI	-	0.107		
VSC-CRI	2.005 (1.178–3.411)	0.010	2.205 (0.444–10.951)	0.334
SII_Postop-SII_Preop	-	0.927		
SII_Day 1-SII_Preop	-	0.709		
SII_Day 1-SII_Postop	-	0.731		
SIRI_Postop-SIRI_Preop	-	0.211		
SIRI_Day 1-SIRI_Preop	1.087 (1.005–1.176)	0.036		
SIRI_Day 1-SIRI_Postop	-	0.091		
AISI_Postop-AISI_Preop	-	0.895		
AISI_Day 1_AISI_Preop	-	0.224		
AISI_Day 1-AISI_Postop	-	0.088		
NLR_Postop-NLR_Preop	-	0.371		
NLR_Day 1-NLR_Preop	-	0.691		
NLR_Day 1-NLR_Postop	-	0.747		
MLR_Postop-MLR_Preop	-	0.384		
MLR_Day 1-MLR_Preop	-	0.977		
MLR_Day 1-MLR_Postop	-	0.543		
PLR_Postop-PLR_Preop	-	0.757		
PLR_Day 1-PLR_Preop	-	0.219		
PLR_Day 1-PLR_Postop	-	0.275		

Abbreviations: AISI_Day1, aggregate index of systemic inflammation from day one after surgery; AISI_Preop, preoperative aggregate index of systemic inflammation; AISI_Postop, postoperative aggregate index of systemic inflammation; CI, confidence interval; Clear_Creat_Preop, preoperative creatinine clearance; Leu_Day1, leukocytes count from day one after surgery; Leu_Preo, preoperative leukocytes count; Leu_Postop, postoperative leukocytes count; Lf_Day1, lymphocytes from day one after surgery; Lf_Preop, preoperative lymphocytes; Lf_Postop, postoperative lymphocytes; M_Day1, monocytes count from day one after surgery; M_Postop, postoperative monocytes count; M_Preop, preoperative monocytes count; MLR_Day1, monocyte-to-lymphocyte ratio from day one after surgery; MLR_Postop, postoperative monocyte-to-lymphocyte ratio; MLR_Preop, preoperative monocyte-to-lymphocyte ratio; MPV_Day1, mean platelet volume from day one after surgery; MPV_Preop, preoperative mean platelet volume, MPV_Postop, postoperative mean platelet volume; MV, mechanical ventilation; N_Day1, neutrophils count from day one after surgery; N_Preop, preoperative neutrophils count; N_Postop, postoperative neutrophils count; NLR_Day1, neutrophil-to-lymphocyte ratio from day one after surgery; NLR_Preop, preoperative neutrophil-to-lymphocyte ratio; NLR_Postop, postoperative neutrophil-to-lymphocyte ratio; *p* probability value; OR, odds ratio; *p* probability value; P_Day1, platelet count from day one after surgery; P_Preop, preoperative platelet count; P_Postop, postoperative platelet count; PLR_Day1, platelet-to-lymphocyte ratio from day one after surgery; PLR_Preop, preoperative platelet-to-lymphocyte ratio; PLR_Postop, postoperative platelet-to-lymphocyte ratio; RCRI, Revised Cardiac Risk Index; SII_Day1, systemic inflammatory index from day one after surgery; SII_Preop, preoperative systemic inflammatory index; SII_Postop, postoperative systemic inflammatory index; SIRI_Day1, systemic inflammatory response index from day one after surgery; SIRI_Preop, systemic inflammatory response index; SIRI_Postop, postoperative systemic inflammatory response index; VSC-CRI, Vascular Surgery Group Cardiac Risk Index.

**Table 5 jcm-13-06338-t005:** Characteristics of the ROC curve of the studied variables in predicting in-hospital death (*n* = 114).

Variable	ROC			Cut Off	Sensitivity (%)	Specificity (%)
	AUC	*p*	CI 95%			
VSG-CRI	0.864	0.001	0.748–0.981	6.5	80.0	85.3
RCRI	0.698	0.057	0.494–0.902	-	-	-
NLR_Preop	0.642	0.149	0.449–0.835	-	-	-
SIRI_Preop	0.677	0.084	0.476–0.878	-	-	-
SII_Preop	0.613	0.334	0.384–0.842	-	-	-
AISI_Preop	0.646	0.240	0.402–0.889	-	-	-
PLR_Preop	0.534	0.783	0.292–0.776	-	-	-
MLR_Preop	0.615	0.413	0.340–0.889	-	-	-
NLR_Postop	0.758	0.006	0.575–0.941	6.53	80.0	64.2
SIRI_Postop	0.716	0.016	0.540–0.891	2.57	100	45.0
SII_Postop	0.692	0.032	0.517–0.867	949.97	100	45.0
AISI_Postop	0.661	0.196	0.417–0.904	-	-	-
PLR_Postop	0.521	0.863	0.281–0.761	-	-	-
MLR_Postop	0.633	0.231	0.415–0.851	-	-	-
NLR_Day1	0.620	0.444	0.313–0.928	-	-	-
SIRI_Day1	0.664	0.286	0.362–0.966	-	-	-
SII_Day1	0.571	0.629	0.284–0.857	-	-	-
AISI_Day1	0.600	0.504	0.306–0.894	-	-	-
PLR_Day1	0.389	0.283	0.186–0.592	-	-	-
MLR_Day1	0.589	0.537	0.306–0.872	-	-	-
SII_Postop-SII_Preop	0.631	0.030	0.512–0.750	414.43	100	51.4
SII_Day 1-SII_Preop	0.492	0.946	0.251–0.733	-	-	-
SII_Day 1-SII_Postop	0.437	0.608	0.195–0.679	-	-	-
SIRI_Postop-SIRI_Preop	0.712	0.020	0.534–0.890	2.79	80.0	66.1
SIRI_Day 1-SIRI_Preop	0.659	0.301	0.358–0.959	-	-	-
SIRI_Day 1-SIRI_Postop	0.558	0.696	0.268–0.848	-	-	-
AISI_Postop-AISI_Preop	0.593	0.317	0.411–0.774	-	-	-
AISI_Day 1_AISI_Preop	0.589	0.532	0.310–0.868	-	-	-
AISI_Day 1-AISI_Postop	0.525	0.871	0.225–0.825	-	-	-
NLR_Postop-NLR_Preop	0.725	0.004	0.571–0.878	2.54	100	45.9
NLR_Day 1-NLR_Preop	0.578	0.594	0.292–0.864			
NLR_Day 1-NLR_Postop	0.442	0.602	0.225–0.659			
MLR_Postop-MLR_Preop	0.594	0.445	0.452–0.837			
MLR_Day 1-MLR_Preop	0.536	0.806	0.250–0.821			
MLR_Day 1-MLR_Postop	0.394	0.358	0.170–0.619			
PLR_Postop-PLR_Preop	0.550	0.470	0.414–0.687			
PLR_Day 1-PLR_Preop	0.305	0.109	0.065–0.544			
PLR_Day 1-PLR_Postop	0.308	0.113	0.071–0.545			

Abbreviations: AISI_Day1, aggregate index of systemic inflammation from day one after surgery; AISI_Preop, preoperative aggregate index of systemic inflammation; AISI_Postop, postoperative aggregate index of systemic inflammation; AUC, area under the curve; CI, confidence interval; MLR_Day1, monocyte-to-lymphocyte ratio from day one after surgery; MLR_Postop, postoperative monocyte-to-lymphocyte ratio; MLR_Preop, preoperative monocyte-to-lymphocyte ratio; NLR_Day1, neutrophil-to-lymphocyte ratio from day one after surgery; NLR_Preop, preoperative neutrophil-to-lymphocyte ratio; NLR_Postop, postoperative neutrophil-to-lymphocyte ratio; *p* probability value; PLR_Day1, platelet-to-lymphocyte ratio from day one after surgery; PLR_Preop, preoperative platelet-to-lymphocyte ratio; PLR_Postop, postoperative platelet-to-lymphocyte ratio; RCRI, Revised Cardiac Risk Index; SII_Day1, systemic inflammatory index from day one after surgery; SII_Preop, preoperative systemic inflammatory index; SII_Postop, postoperative systemic inflammatory index; SIRI_Day1, systemic inflammatory response index from day one after surgery; SIRI_Preop, systemic inflammatory response index; SIRI_Postop, postoperative systemic inflammatory response index; VSC-CRI, Vascular Surgery Group Cardiac Risk Index.

## Data Availability

They may be available on request via correspondent author email.

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
