# Peer review of "The Predictive Value of Perioperative Inflammatory Indexes in Major Arterial Surgical Revascularization from Leriche Syndrome"

_jcm, 2024, doi:10.3390/jcm13216338_

Round 1
Reviewer 1 Report
Comments and Suggestions for Authors
Thank you for submitting your work. I enjoyed reading the paper. From a biostatistical point of view, you manuscript carries a significant methodological flaw: that is, comparing data from 5 patients to data from 109 patients. The paper also does not introduce any novelty.
Comments on the Quality of English LanguageThe language used needs minor editing. I recommend proof-reading by a native speaker.
Author Response
Dear Professor,
We are submitting the revised version of our manuscript entitled “The predictive value of perioperative inflammatory indexes in major arterial surgical revascularization from Leriche syndrome” (jcm-3219357) to be considered for publication in Journal of Clinical Medicine as an Article. The initial version of our manuscript has been revised taking into consideration your comments.
Each point made by you is addressed below separately, explaining what we have changed and where the changes are to be found in the manuscript (the changes are highlighted in the revised manuscript).
We thank you for your useful suggestions and comments and your kind words about our manuscript. We greatly appreciate the time and efforts you have taken for the evaluation of our work!
Comments and Suggestions for Authors
Thank you for submitting your work. I enjoyed reading the paper. From a biostatistical point of view, you manuscript carries a significant methodological flaw: that is, comparing data from 5 patients to data from 109 patients. The paper also does not introduce any novelty.
Comments on the Quality of English Language
The language used needs minor editing. I recommend proof-reading by a native speaker.
Response:
Thank you for your observations! We revised the manuscript including references. We also did some minor editing changes. All these are highlighted in the revised manuscript.
In our study population of 114 patients, we have identified five non-survivors, representing a mortality rate of 4.38%. This rate aligns with the literature on major arterial surgical revascularization for Leriche syndrome. Indeed, there is a significant difference between the number of survivors and non-survivors. To address this issue, we primarily used nonparametric statistical methods since most of the data did not follow a normal distribution. We explicitly mentioned this limitation in the study presentation.
We restructured the Introduction to make it more readable. Additionally, we completed the study of in-hospital mortality by analyzing the delta (difference) values of SII, SIRI, AISI, NLR, MLR, and PLR. We also analyzed the inflammatory indexes within each group at three perioperative moments: preoperative, immediately postoperative, and one day after surgery. We believe that the changes we made have added value to our manuscript.
Dear Reviewer, we hope that the current version of the manuscript appropriately addresses your suggestions.
Thank you for kindly considering our manuscript!
Looking forward to your decision!
Sincerely Yours,
Anca Drăgan

Reviewer 2 Report
Comments and Suggestions for Authors
Dear editor,
I reviewed the article entitled “The predictive value of perioperative inflammatory indexes in major arterial surgical revascularization from Leriche syndrome”. My comments are below.
1- Introduction section is not acceptable. It is so complex and confused. It should be written systematically. In present form, it is not acceptable.
Please write the introduction as follows: Leriche sydnropme, what and therapy; after thatn the role of inflammation in Leriche, after than the ourtcome of studies and last the aim.
In the current form, they are so coınfused. Authors state the aim of the study and then, they again and again exoplain the infllamtory markers??? It is not suitable
2- Table 1, what is clear creat??
3- Table 2, RCRI and VSG-CRI scores should be compared with the survivors. Why these scores did not compare between two groups? Is there any specific reason?
4- The delta value (difference) of SII, SIRI, PLR, MLR NLR and AISI should be compared between the survivors and non survivors.
5- Did the authors compared the SII, SIRI, PLR, MLR NLR and AISI in survivors and non survıvors? Intragroup comparison (repeated measure ANOVA and Friedman)? Please perform and help from a statistician. Performed analyses are not enough.
6- Figure 2, thw which patients? Survivors or non survivors or all group? Please present the survıvırs and non survivors in the sam figure with the different color.
Sincerely.
Comments on the Quality of English Languageminor changes required.
Author Response
Dear Professor,
We are submitting the revised version of our manuscript entitled “The predictive value of perioperative inflammatory indexes in major arterial surgical revascularization from Leriche syndrome” (jcm-3219357) to be considered for publication in Journal of Clinical Medicine as an Article. The initial version of our manuscript has been revised taking into consideration your comments.
Each point made by you is addressed below separately, explaining what we have changed and where the changes are to be found in the manuscript (the changes are highlighted in the revised manuscript).
We thank you for your useful suggestions and comments and your kind words about our manuscript. We greatly appreciate the time and efforts you have taken for the evaluation of our work!
Comments and Suggestions for Authors
I reviewed the article entitled “The predictive value of perioperative inflammatory indexes in major arterial surgical revascularization from Leriche syndrome”. My comments are below.
1- Introduction section is not acceptable. It is so complex and confused. It should be written systematically. In present form, it is not acceptable.
Please write the introduction as follows: Leriche sydnropme, what and therapy; after thatn the role of inflammation in Leriche, after than the ourtcome of studies and last the aim.
In the current form, they are so coınfused. Authors state the aim of the study and then, they again and again exoplain the infllamtory markers??? It is not suitable
2- Table 1, what is clear creat??
3- Table 2, RCRI and VSG-CRI scores should be compared with the survivors. Why these scores did not compare between two groups? Is there any specific reason?
4- The delta value (difference) of SII, SIRI, PLR, MLR NLR and AISI should be compared between the survivors and non survivors.
5- Did the authors compared the SII, SIRI, PLR, MLR NLR and AISI in survivors and non survıvors? Intragroup comparison (repeated measure ANOVA and Friedman)? Please perform and help from a statistician. Performed analyses are not enough.
6- Figure 2, thw which patients? Survivors or non survivors or all group? Please present the survıvırs and non survivors in the sam figure with the different color.
Comments on the Quality of English Language
minor changes required.
Response:
Thank you for your observations! We revised the manuscript including references. We also did some minor editing changes. All these are highlighted in the revised manuscript.
- Based on your suggestions, we have rewritten the Introduction. We believe that the new version is more readable.
- We have also clarified that "Clear creat" in Table 1 refers to the preoperative creatinine clearance. We have corrected the abbreviation.
- While some of the information was within the text, we included the data related to the risk scores analysis in the two subgroups in Table 1 to ensure greater accuracy. Non-survivors presented significantly higher VSG-CRI (p=0.005, Mann-Whitney test) compared to survivors. RCRI did not present different values in the two subgroups (p=0.117, Mann-Whitney test).
- We added the analysis of the delta values of SII, SIRI, AISI, NLR, PLR, and MLR to our in-hospital mortality study. The analysis is added in Table 1 (comparison between subgroups), Table 4 (binary logistic regression), and Table 5 (ROC analysis). Additionally, Figure 4 has been consecutively modified.
- We compared the six inflammatory indexes between two subgroups (survivors and non-survivors) as shown in Table 1. Additionally, we included intragroup analysis of the inflammatory indexes at three perioperative time points based on our recommendation (new Table 3). We used Friedman analysis with post-hoc analysis (Wilcoxon signed-rank test with Bonferroni correction) due to the non-normal distribution of the data.
- Based on our suggestion, we modified Figure 2. We graphically (box plot) presented the inflammatory indexes in the three studied perioperative moments (preoperative, immediately postoperative, and day one after surgery) in the two subgroups of patients.
Comments on the Quality of English Language- Based on your suggestions, we made some minor changes in language setting.
Dear Reviewer, we hope that the current version of the manuscript appropriately addresses your suggestions.
Thank you for kindly considering our manuscript!
Looking forward to your decision!
Sincerely Yours,
Anca Drăgan

Round 2
Reviewer 1 Report
Comments and Suggestions for Authors
I congratulate the authors for their extensive revision of the paper.
Reviewer 2 Report
Comments and Suggestions for Authors
authors responded in a satisfactory way. thank you.